# A Study on Emissions from Drayage Trucks in the Port City-Focusing on the Port of Incheon

**Hyangsook Lee [1], Hoang Thai Pham [1], Chihoon Kim [2] and Kangdae Lee [3,\*]**

[1] Graduate School of Logistics, Incheon National University, Incheon 22012, Korea; hslee14@inu.ac.kr (H.L.); hoangphamhp92@gmail.com (H.T.P.)

[2] Seoul Metro Line9 Cooperation, Seoul 07505, Korea; kimchi@metro9.co.kr

[3] Department of Packaging, Yonsei University, Wonju 26403, Korea

\* Correspondence: pimeson@yonsei.ac.kr; Tel.: +82-33-760-2241

**Abstract:** As a result of growing international trade, port-related emission is a spreading issue for urban areas located near ports, especially, hub port cities where population density is concentrated. The awareness of rapidly increasing drayage trucks moving cargo between the port and its hinterland has motivated the necessity of a detailed look at negative environmental impacts of these truck fleets on the achievement of sustainability goals. This study analyzes emission inventory from trucking activities around the Port of Incheon (POI), especially focusing on major air pollutants, and suggests ideas to support establishing new policies in port area. Data on the number of truck, the year of production, the type of fuel, etc. during 2018 were collected from Incheon Port Security and Korea Transport Safety Authority. A bottom-up methodology is applied based on guideline from the U.S. Environmental Protection Agency (EPA). As results, the major role of drayage truck fleets to local air pollution was highlighted with the high contribution of CO and $NO_x$ emissions. Hence, this study suggested the establishment of Emission Control Area (ECA) and Affected Zone on the landside as well as implementing Integrated Information System and Truck Appointment System to reduce congestion at gate, limit the number of emissions and minimize negative impacts to local community.

**Keywords:** urban freight transport; drayage truck; port city; emission; sustainability

## 1. Introduction

In recent decades, urbanization has pushed up population concentration and economic growth in urban cities. High population density has made the cities become center nodes for consumption and production. The flows of urban freight are transported between major land uses as manufacturing sites, warehouses and retail facilities through supply networks. Thus, road-based transportation systems in urban areas are developed and expanded quickly to support increasing unavoidable levels of logistics demand of inhabitants and resident businesses. However, this rapid development and the complexity of freight movement also bring about a range of serious socio-economic and environmental impacts including increasing urban congestion and pollutions (i.e., air, noise). At the city level, these impacts have been affecting severely to local community's health and life quality, preventing cities achieving sustainability, and contributing significantly to thinking that "cities are not safe", then, driving sub-urban relocation trend [1], meanwhile, at a global scale, there are recognized as principal contributors to climate change effects [2].

The consideration about the interaction between transport and urban planning started from the mid-1980s, however, the mainstream of studies primarily targeted on private and public passenger transport in and around cities, instead of showing proper attention on the movement of commodities and services [3]. After that, there has been a growing literature on the urban freight transport with two

main categories: operational models, which focus on flow management improvement, and systematic models that primarily consider the impact of urban logistics modifications on the flows generated [4]. However, the scientific community have paid less attention to an interaction between urban freight transport and sustainability goals in cities [5] and the optimization of urban logistics activities through policy and planning [6]. The big cities, with high population and traffic demand, are the main target of previous studies due to the high potential environmental impact reduction, based on the quantity of transport. Urban freight planning in the small and medium cities also are interesting topic because the transport in those areas is considered to be less efficient than in large cities (less concentrated population, dispersed transport flows), however a large number of small cities together will be expected to generate a higher total potential effect [7].

Freight transport is considered as the main contributor to urban traffic issues, especially congestion. Manners-Bell [8] explained the main reason here is the high usage rate of road and rail network in urban areas. Although freight traffic accounted for around 15% of total urban traffic volumes, it generated huge congestion on key highways [9]. In both road and rail networks, freight and passenger flows interact and affect together. Traffic can be delayed at crossings and complicated on passenger-freight shared routes due to adding of freight flows.

Thanks to increasing understanding of environmental impacts of urban freight flows on city sustainability, local authorities and academic fields recognized the contribution of urban freight transport to air pollutant emissions, not only greenhouse gases but also non-greenhouse gases including particulate matter (PM) and nitrogen oxides ($NO_x$) [10,11]. In Paris, France, in 2017, urban freight transport contributed about 14% of the total vehicle kilometers travel, however, it was responsible for 38% of the total $NO_x$ emission, 43% of $SO_x$ emission and nearly 59% of the total PM emission from transport sector [12]. The percentages in London were 36% and 39% of total $NO_x$ and PM emission, respectively, as of 2014 [13]. Diesel vehicles (i.e., truck, train, ship) were considered as the main source of particulate emission inside most cities [9,14]. The main component of freight transportation is short-haul drayage trucks, which deliver cargoes inside ports and between ports to distribution facilities or warehouses. This trucking activity also has remarkable waiting time (idle, creep) at port gates for cargo or container handling process. In addition, hub port cities, focal points of international trade and traffic, tend to be located in metropolitan areas, in other words, densely populated areas. More truck flows are generated there, and then, they amplify the scale and complexity of those impacts on local areas. Therefore, the need for understanding correctly the close link between urban air pollution and freight trucking activities in port areas has been raised.

The Korean government planned 10-year comprehensive plans and managed integrated air management system to improve air quality. A national emission inventory, Clean Air Policy Support System (CAPSS) has been developed, however, it also is unreliable due to inconsistency with other academic studies. Another problem with this system is that it paid attention to $CO_2$ emission [15]. CAPSS extended the scope of emission contents by considering other air pollutants including PM emission, however, data is updated too late and sources of the applied emission factors are not clear. The Korea government released a new Comprehensive Plan of Particular Matter Management [16] on September 26th, 2018, announcing a new target of 30% PM emission reduction until 2022, instead of concerning only $CO_2$ as before. Therefore, it raised a research demand about the freight trucking emission in port areas to investigate whether trucking activities contribute significantly to local air pollution and how much of this contribution. Hence it can be used as a reliable and up-to-date source for the green policy decision-making process, especially focusing on PM emission reduction goals.

The Port of Incheon (POI) is located in the third biggest metropolitan cities in Korea. It also plays an important role in international trade as a gateway to the north-west region. As a result, local authority in Incheon, home of the second largest port in Korea, has to face a dilemma: on the positive side, international trade and increasing logistics activities will offer more jobs and contribute actively to local economic development and competitiveness. However, larger negative impacts also will come together with benefits and affect seriously to life quality of local citizen. Unfortunately, trucking emission in

port areas received little consideration from academic researchers. Han et al. [17] considered drayage truck emission as a part of total emissions in POI. However, a top-down methodology and old data are applied, thus, it provided less reliable results. In addition, the emission emitted in affected-zone was ignored in the study.

Therefore, this paper aims to clarify the contribution of trucking activities in the Incheon port area to urban air pollution, then, suggest and discuss possible policies that might help cut down serious emissions. To do this, the paper firstly estimates emission inventory from trucking activities in and outside port gates. Next, annual paved road particulate emission, caused mainly by road traffic, is calculated promptly. The combined emission inventory will improve the understanding of freight activities-city air pollution relation, and motivate the promulgation of fitting public policies and activities to improve air quality, then, assist local authorities, academics, and planners address the urban sustainability problems.

## 2. Literature Review

### 2.1. Emission Estimation Models

Capturing emissions from road freight has received uninterrupted attention from researchers and public authorities since the middle of the 1980s. A rich stream of emission estimation models has been introduced and classified into macroscopic models, which apply average inputs for wide-area assessment. On the other hand, microscopic models consider particular case studies with hot-stabilized operation at a specific point of time.

In terms of macroscopic models, Hickman et al. [18], in a project report called Methodologies for Estimating Air Pollutant Emissions from Transport (MEET), introduced a model that use actual on-road measured parameters to capture energy consumption and emissions by heavy goods vehicles. In Europe, the European Environment Agency has published a series of technical guidebook [19] to prepare national emission inventory since 1996. In the guidebooks, Computer Programme to Calculate Emission from Road Transportation (COPERT) which applied speed regression models for emissions estimation has been suggested for capturing emissions released from drayage trucks. COPERT has been applied widely for various research purposes, vehicle classes, engine classifications, emission standards and fuel types. In the United States (U.S.), the California Air Resources Board [20] applied Emission Factor Model (EMFAC) which considers vehicle average speed, fuel type and environmental conditions to figure out the number of emissions. On the other hand, at the U.S. national level, the U.S. Environment Protection Agency (EPA), an official organization from the U.S. Government, developed MOBILE and the Motor Vehicle Emission Simulator (MOVES) models to capture drayage truck emission within U.S. boundary [21,22]. In addition, there are abundant of similar models such as Handbook Emission Factors for Road Transport (HBEFA), Greenhouse Gases, Regulated Emissions, and Energy Use in Transportation (GREET), Lifecycle Emissions Model (LEM), VERSIT plus–light-duty (VERSIT+ LD) and International Vehicle Emission (IVE) [23].

In terms of the microscopic aspect, Instantaneous Fuel Consumption Model (IFCM) which considers specific vehicle characteristics and environment parameters were described by Bowyer et al. [24]. Jimenez-Palacios [25] presented Vehicle Specific Power (VSP) model which mainly depends on the change of second-by-second emissions released from diesel trucks. Physical Emission Rate Estimator (PERE) model was developed by Nam and Giannelli [26] to support for MOVES model that allows users to modify inputs. Based on engine power demand and actual speed, Passenger Car and Heavy Duty Emission Model (PHEM) can simulate fuel consumption of a range of vehicles and engines, especially for passenger car and heavy-duty truck [27]. Comprehensive Modal Emission Model (CMEM) was designed for heavy goods vehicles based on second-by-second tailpipe emission data [28]. Its reliability highly depends on the accuracy of specific engine speed and friction coefficient.

The accuracy and application range of models have been improved remarkably in recent years. Models were developed to apply broadly in different study fields, then, applied to relative case

studies properly. Among above-mentioned models, COPERT and CMEM are the most frequently used models in macroscopic and microscopic aspects, respectively, to capture emission from road freight transportation [23]. Both models work well with various vehicles, however, the accuracy of the results highly relies on the quality of input parameters.

*2.2. Emission Reduction Solutions*

2.2.1. Truck Appointment System

Quantity of vehicle and operating time including idle time are main factors that determine the amount of emission released, therefore, amount of emission reduced will be proportional to the number of trucks decreased or idle time cut down. To reduce emissions but still retain economic efficiency, a system called Truck appointment system (TAS) in U.S. or Vehicle Booking System (VBS) internationally was suggested for decreasing free-truck trips, then, reducing the amount of emission released. TAS defines time windows when trucks can perform the service like loading and unloading containers at port. Thus, terminal operators can control the number of trucks entering and operating per zone to match the capacity of terminal throughout the day [29,30]. First of all, truck carrier needs a user name and password to access to TAS system and book a slot. A slot is necessary to book several hours before arrival with required information about a target container. To ensure impartiality of access for all truck companies, each company just can book only one slot within a certain period of time. Then, a truck driver arrives the port with a received code within the booked appointment time and goes directly to the identified terminal. There are several existing TASs in practice: Truck licensing system at Port Metro in Vancouver, eModal at APL Terminal Pier 300 in Los Angeles, VBS in Port Botany in Sydney and Australia, and Port of Southampton in the United Kingdom [30].

2.2.2. Urban Freight by Rail Network

Another environment-friendly alternative to help reduce emissions at the port area is using existing urban rail transportation network. Railway container transport can connect the seaport with the hinterland region, which can help reduce congestion, emission, and traffic in urban area [31]. The rail can save 50% of energy consumption (it means 50% amount of emissions released) for each tonne-km compared to road freight haulage [32]. In addition, rail enables higher axle weights and volume capacity for lighter weight products. In term of external cost, in the European Union, the road freight transportation performed higher cost per tonne-km than rail freight haulage up to five times [33].

Optimization/simulation models have been purposed to investigate the possibility of a shared system for both cargo and passenger transport. Fatnassi et al. [34] suggested an urban intermodal road-rail transport system combining freight and passenger on-demand rapid transit. A discrete event simulation model was used to analyze the current situation, evaluate market competitive and design a system of moving urban freight by rail in the City of Newcastle [35]. Gonzalez-Feliu applied a socio-economic cost-benefit analysis to assess the suitableness of urban rail logistics in the Parisian region [36]. Nuzzolo et al. [37] suggested using existing railway system and old passenger trains for delivering goods between Naples and Sorrento during off-peak periods.

In addition, with the development of technology, the open opportunities for using underground space for urban freight transportation has been received interests from researchers [38,39]. From the begins of the 1990s, a dedicated underground goods distribution system based on the existing passenger rail network was considered to overcome traffic congestion and environmental issues in Tokyo [40]. CargoCap is an advanced concept of the cargo transportation system, that uses automated electric vehicles—the Caps—travel through underground pipelines, was suggested by Dietrich Stein in 2002 [41]. Cargo Sous Terrain (CST) is an overall logistics system connecting production sites, logistics locations and urban consumers among Swiss cities by using autonomous electric vehicles to transport small-scale cargos [42].

Beside previous research studies, several urban rail transportation cases have been applied successfully in practice. CarGo Tram is a project using 4 km tram line to connect Volkswagen's production site in the city center with its logistics warehouses located near rail terminal for transporting automotive parts and modules in Dresden, Germany. With up to 10 trips per day, this line can deliver 300,000 tons of products annually, thus, it not only can meet demand for daily production and reduce company's warehouse space but also cut-down significantly polluted emissions from trucking activity [31].

In France, Société Nationale des Chemins de Fer Français, a France's national state-owned railway, and supermarket chain Monoprix cooperate to use commuter line D (RER D) to deliver a small part of daily goods, including soft drinks, textiles, beauty and home products in a single train from Monoprix depots located outside the city of Paris to Barcy station inside Paris, then goods are distributed by trucks to several supermarkets of this chain. It results in a reduction of 10,000 trucks entering the urban area with 280 tons in $CO_2$ and 19 tons in $NO_x$ annually [43]. Another similar project, named TramFret, also are operated in Paris by another state-owned public transport operator by using recycled tramway rolling stock. Both cases are operated on the same existing railways with passenger trains by taking advantages of gaps in passenger train operating schedules. Therefore, it requires a high level of integrated information system to arrange a suitable timetable for both freight and passenger trains. A two mixed integer linear model [43] was developed to resolve complicated problems in operating such as multiple deliveries and stations, different delivery times imposed by several clients and different types and size of transported products.

## 3. Materials and Methodologies

### 3.1. Research Scope

POI is composed of five key component ports: North Port, Inner Port, Coastal Port, South Port and New Port. In addition, there are three smaller specialized ports located around the five big ones: Geocheom-do Port, Song-do Port, and Yeongheung-do Port. North Port, with 17 berths, is specialized in handing industrial raw materials such as timber and steel. By maintaining calm water level with lock gate, Inner Port is available for handling semi-conductor equipment, vehicles, and machine parts. South Port and New Port have been developed for handling containers [44]. Inner Port and Coastal Port will be counted as one due to sharing same gates. The study covered the all emissions from trucking activities inside ports, as well as the affected zone (within 0.5 km from port gate), following the guideline from the U.S. Environmental Protection Agency (EPA) [45]. The sulfur content in fuel is set as 15 ppm based on EPA Tier 4 Emission Standard [46]. Emission inventory of 8 target pollutants is defined based on CAPSS system [47]: CO, NOx, SOx, PM, PM10, PM2.5, volatile organic compounds (VOC), and ammonia (NH3). Truck emission during 2018 is calculated according to truck type and port. The geographical visualized research area is showed in Figure 1.

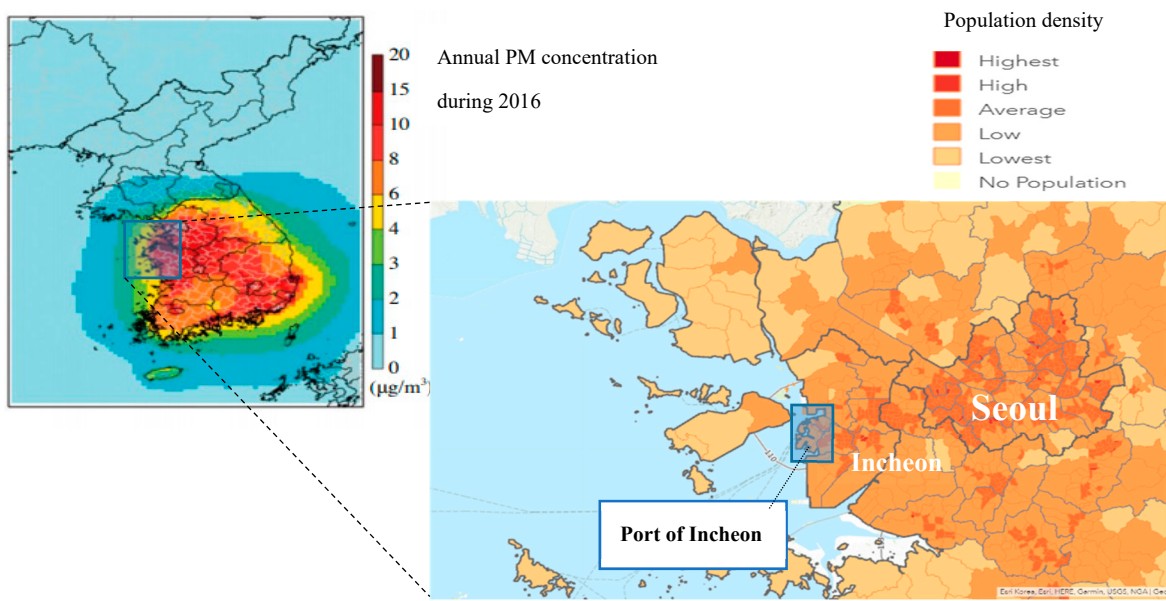

**Figure 1.** Geographical visualized emission and population density of Port of Incheon area. Source: [48] and ArcGIS Online.

*3.2. Data*

Truck activity data are collected from Incheon Port Security. A total of 2,907,990 drayage trucks entered port during 2018 was recorded at port gates and berths, as shown in Table 1. Data from Coastal Port is counted into Inner Port's number because they share gates together. Drayage trucks operated are classified into four main groups: Small truck (less than 1 ton of goods), medium truck (carrying 1–5 tons of goods), large truck (more than 5 tons of goods), and container truck.

**Table 1.** Number of drayage truck by ports during 2018 (unit: vehicle).

| Port | Small Trucks | Medium Trucks | Large Trucks | Container Trucks | Total |
|------|-------------|---------------|--------------|------------------|-------|
| North Port | 8676 | 10,412 | 154,439 | - | 173,527 |
| Inner Port | 44,870 | 53,604 | 786,342 | 162,167 | 1,046,983 |
| South Port | 5625 | 6720 | 98,577 | 529,942 | 640,864 |
| New Port | 59 | 71 | 1035 | 833,109 | 834,274 |
| Others | 10,768 | 12,864 | 188,710 | - | 212,342 |
| Total | 69,998 | 83,671 | 1,229,103 | 1,525,218 | 2,907,990 |

Vehicle data like the year of vehicle production and type of fuel (diesel and LPG) are obtained from the Korea Transport Safety Authority and matched with truck activity data above. Emission standards EURO 0~VI are defined based on year of vehicle production. However, this estimation is not exact completely due to the long gap and complication of the point of new approvals (often in midyear) and enforcements in practice (in most cases 1 year later), as well as the big different in time of approvals of each kind of vehicle (light-duty vehicle, heavy-duty vehicle). After that, based on EURO emission standards, type of fuel, and other vehicle information, appropriate emission factor is decided in the estimation process below.

Among recorded 2,907,990 trucks, 21% of them were manufactured before 2006 with low-technology engines (equivalent to the emission standard EURO 0-III), which release more $NO_x$ and CO emissions. Over half of trucks were made from 2006 to 2013 (equivalent to emission standard EURO IV-V), accounting for 67%. The proportion of new vehicles manufactured after 2013 (with emission standard EURO VI) was just 23% of the total number of trucks. Table 2 lists the number of drayage truck operated in POI during 2018.

**Table 2.** Number of drayage truck by manufacturing year (unit: vehicle).

| Port | Before 2006 (Equivalent to EURO 0-III) | 2006–2013 (Equivalent to EURO IV-V) | After 2013 (Equivalent to EURO VI) | Total |
|---|---|---|---|---|
| North Port | 36,444 | 97,172 | 39,911 | 173,527 |
| Inner Port | 219,866 | 586,311 | 240,806 | 1,046,983 |
| South Port | 134,582 | 358,886 | 147,396 | 640,864 |
| New Port | 175,199 | 467,190 | 191,885 | 834,274 |
| Others | 44,594 | 118,909 | 48,839 | 212,342 |
| Total | 610,685 | 1,628,468 | 668,837 | 2,907,990 |
| % | 21 | 56 | 23 | 100 |

In terms of fuel used, over 90% of drayage trucks operated in POI during 2018 used diesel. The quantity of LPG vehicle is still limited at 6%. Number of drayage truck operated by fuel types is shown in Table 3.

**Table 3.** Number of drayage truck by fuel types (unit: vehicle).

| Port | Diesel | LPG | Total |
|---|---|---|---|
| North Port | 163,115 | 10,412 | 173,527 |
| Inner Port | 984,162 | 62,821 | 1,046,983 |
| South Port | 602,412 | 38,452 | 640,864 |
| New Port | 784,217 | 50,057 | 834,274 |
| Others | 199,601 | 12,741 | 212,342 |
| Total | 2,733,507 | 174,483 | 2,907,990 |
| % | 94 | 6 | 100 |

Data on traveling distance at each port is defined as an average traveling distance from berths to their respective gates at that port, as shown in Table 4. The annual meteorological data on daily maximum and minimum temperature, average wind speed, average precipitation during 2018 in POI were collected from the official portal of the Korean government.

**Table 4.** Average travelling distances by ports (unit: km).

| Port | Average Travelling Distances |
|---|---|
| North Port | 0.37 |
| Inner Port | 0.914 |
| South Port | 0.349 |
| New Port | 0.427 |
| Others | 0.713 |

*3.3. Calculation Process*

The total emissions from trucking activities in the port area are aggregated from 2 main sources: drayage truck emissions and paved road PM emission. For drayage truck emissions, COPERT software is applied to capture emissions released during engine combustion in both start and normal operation processes. Then, paved road dust emissions caused by daily traffic on the road is estimated by referring guidance from EPA.

### 3.3.1. Emissions from Drayage Trucks

Total emissions from a truck are estimated complexly from a combination of exhaust and non-exhaust pollutants estimation. Non-exhaust part considers fuel evaporation and dust from tire and brake attrition while exhaust part focuses on gas and PM emissions from the engine.

$$E_{truck} = \sum E_{exh} + E_{evap} + E_{att},$$  (1)

where,

$E_{exhaust}$ is the number of exhausted emissions,
$E_{evap}$ is the number of emissions from fuel evaporation, and
$E_{att}$ is the number of PM emissions emitted from tire and brake attrition.

Exhaust Emission

Exhaust emissions are added up from 2 parts, hot part and cold part. Hot part happens when the engine $k$ and after-treatment system are operated stability and reach their normal operating temperature. The other part, cold part, covers emissions from engine $k$ before it reaches the normal temperature. The calculation process is applied concurrently for all target pollutants.

$$E_{exhaust} = \sum_{k,r} E_{hot\ i,k,r} + E_{cold\ i,k,r},$$  (2)

Emission at normal operation is estimated by the following equation:

$$E_{hot\ i,k,r} = ef_{hot\ i,k,r} \times D_k = ef_{hot\ i,k,r} \times N_k \times M_{k,r}$$  (3)

$$ef_{hot\ i,k,r} = \frac{\alpha + \gamma v + \varepsilon v^2 + \zeta v^2}{1 + \beta v + \delta v^2} \times \left(1 - RF_{k,r}\right)$$  (4)

where,

$ef$ is the emission factor (g/km),
$v$ is the average speed of the vehicle (km/h),
coefficients $\alpha$ to $\zeta$ are empirically derived for each engine to calculate for emission factor based on actual speed of the vehicle,
$RF$ is the reduction factor,
$D$ is total traveled distance (km),
$N$ is the number of the vehicle (veh),
$M$ is mileage per vehicle (km/veh),
*hot* is stabilized (hot) engine operation,
$k$ is *the* type of engine,
$i$ is *a* pollutant, and
$r$ is road class (urban, rural, highway).

Based on emission standard inputted, the software determines age and type of engine used in vehicle, then, decides coefficients $\alpha$ to $\zeta$ to estimate proper emission factor applied in Equation (3).

Emission at cold part is estimated by following equation:

$$E_{cold\ i,k,r} = \beta_{i,k,r} \times L_k \times ef_{hot\ i,k,r} \times \left(\frac{ef_{cold}}{ef_{hot}} - 1\right) = \beta_{i,k,r} \times N_k \times M_k \times ef_{hot\ i,k,r} \times \left(\frac{ef_{cold}}{ef_{hot}} - 1\right)$$  (5)

where,

*β* is a fraction of mileage driven with a cold engine or the catalyst operated below the light-off temperature,
*L* is the total trip length (km),
*N* is the number of vehicle (veh),
*M* is mileage per vehicle (km/veh),
$ef_{cold}/ef_{hot}$ is cold/hot emission quotient, defined by:

$$ef_{cold}/ef_{hot} = l - m \times temp \tag{6}$$

where,
*temp* is ambient temperature, and
$l = 1.47$, $m = 0.009$ (if fuel is gasoline), $l = 1.34$, $m = 0.008$ (if fuel is diesel).

Emission from Fuel Evaporation

The first main source of non-exhaust emissions is fuel losses by evaporation in the fuel system of vehicles that use gasoline fuel due to the relatively high vapor pressure of gasoline. It happens both in parking time and operating time. In COPERT, fuel losses from evaporation are estimated from 3 phases: diurnally (temperature change during the day—$E_{diurnal}$), hot soak (hot fuel was kept in the reservoir after the operation—$E_{soak}$) and running losses (warm fuel ran back to the reservoir during operation—$E_{running}$), as expressed below:

$$E_{evap\ i} = E_{diurnal\ i} + E_{soak\ i} + E_{running\ i} \tag{7}$$

PM Emission from Tire and Brake Attrition

PM emission from tire and brake attrition is another main source of non-exhaust emissions. Most of them become airborne and affect primarily to ambient PM concentrations in air. COPERT does not consider dust from resuspension of loose material on the road surface. For estimating the amount of direct PM emission from trucks, a similar equation to Equation (3) is suggested by considering traveled distance in km (*D*) and appropriate average emission factor, as follows.

$$E_{att} = D \times ef_{PM} \tag{8}$$

3.3.2. Particulate Emission from Abrasion of Paved Road

Re-suspended PM emission is originated from loose material that already exists on the surface of paved road. In addition, the annual amount of emissions also can be mitigated by natural factors as rain. Applying precipitation correction term, long-term average dust emission can be estimated by the following equation:

$$E_{PM\ road} = EF_{PM\ road} \times VKT \times \left(1 - \frac{P}{4N}\right) \tag{9}$$

where,

*E* is the amount of PM emission (tons),
*EF* is the emission factor (g/km),
*P* is number of wet days with at least 0.254 mm (0.01 in) of precipitation during the averaging period (days),
*N* is the number of days in the averaging period (=365 days),
*VKT* is Vehicle Kilometers of Traveled (km).

Emission factor value varies with something called silt loading (sL) of the road surface and average weight (tons) of total vehicles traveling on the road (W). Silt loading presents the mass of

silt-size material (dust that is not larger than 75 micrometers), expressed in grams per square meter. EPA [49] suggested equation for estimating emission factor as below:

$$EF_{PM\ road} = k(sL)^{0.91}(W)^{1.02} \tag{10}$$

where, $k$ is particulate size multiplier ($k$ = 3.23 for TSP, 0.62 for PM 10 and 0.15 for PM 2.5), expressed in g/VKT.

After emission released from drayage truck ($E_{truck}$) and re-suspended PM emission were estimated concurrently by air pollutant types, the total emission inventory was summarized.

## 4. Analysis Results

The total amount of drayage trucking emissions in POI during 2018 is 5348.7 tons including 965.7 tons of CO, 3686.9 tons of $NO_x$, 5.8 tons of $SO_x$, 258.4 tons TSP (191.6 tons of PM10 and 122.8 tons of PM 2.5), 108.9 tons of VOC and 8.7 tons of $NH_3$. $NO_x$ and CO are the two principal pollution sources in POI, accounting for 69% and 18% of the total amount of emissions, while the contribution of $SO_x$ and $NH_3$ to port air pollution were relatively minor. Emissions from ports are shown in Figure 2 below.

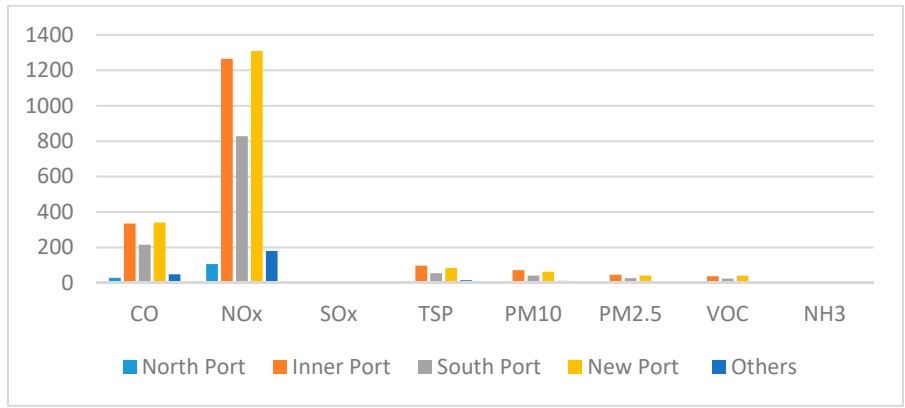

**Figure 2.** Ports comparison by air pollutants (unit: ton).

Table 5 presents trucking emission by ports including four main ports, North Port, Inner Port, South Port, and New Port, and three other specialized ports. Inner Port and New Port shared a similar amount of emissions as the most polluted ports. South Port showed the third-largest share of the total emissions. North Port released the smallest amount of emissions.

**Table 5.** Truck emission inventory by the port in POI during 2018 (unit: ton).

| Port | CO | $NO_x$ | $SO_x$ | TSP | PM10 | PM2.5 | VOC | $NH_3$ |
|------|------|--------|--------|------|------|-------|------|--------|
| North Port | 28.2 | 105.8 | 0.2 | 8.6 | 6.4 | 4.0 | 3.2 | 0.3 |
| Inner Port | 334.6 | 1264.3 | 2.0 | 96.5 | 71.4 | 44.9 | 37.4 | 3.4 |
| South Port | 215.3 | 827.9 | 1.3 | 54.4 | 40.4 | 26.3 | 23.5 | 1.8 |
| New Port | 339.5 | 1308.6 | 2.0 | 84.2 | 62.6 | 40.9 | 39.4 | 2.7 |
| Others | 48.1 | 180.3 | 0.3 | 14.7 | 10.9 | 6.7 | 5.4 | 0.5 |
| Total | 965.7 | 3686.9 | 5.8 | 258.4 | 191.6 | 122.8 | 108.9 | 8.7 |

Table 6 illustrates the trucking emission inventory by two drayage truck groups in POI during 2018. It is interesting that large and container truck dominated the total amount of emissions in POI. It contributed 99.5% of CO, 99.5% of $NO_x$, 99.5% of $SO_x$, 99.2% of TSP, 97.3% of VOC, and 99.1% of $NH_3$. In contrast, small and medium drayage trucks just took a small proportion of the total amount of emissions. The reason here is the number of large drayage truck and container is accounted for 94.7% total truck in POI.

**Table 6.** Drayage truck emission inventory by vehicle type in POI during 2018 (unit: ton).

| Type of vehicle | CO | $NO_x$ | $SO_x$ | TSP | PM10 | PM2.5 | VOC | $NH_3$ |
|---|---|---|---|---|---|---|---|---|
| Small/Medium truck | 4.4 | 17.2 | 0.0 | 2.2 | 1.5 | 0.9 | 3.0 | 0.1 |
| Large/Container truck | 961.3 | 3669.8 | 5.7 | 282.5 | 195.2 | 123.1 | 105.9 | 8.6 |
| Total | 965.7 | 3686.9 | 5.8 | 258.4 | 191.6 | 122.8 | 108.9 | 8.7 |

Comparing to an emission inventory in Incheon during 2016 declared by Korea National Air Pollutants Emission Service [50], the amount of $NO_x$ emission released from drayage trucks in POI during 2018 accounted for 7.5% of the total $NO_x$ emission in Incheon area. Drayage truck in the port area also released 2.55% of total CO emission and 1.36% of total TSP emission in Incheon. The amount of $SO_x$, VOC, and $NH_3$ emitted from the truck are minuscule (less than 0.2%) when comparing to the total amount of Incheon. This comparison highlights the significant contribution of drayage truck in term of $NO_x$ and CO emission to Incheon's air pollution.

## 5. Discussion and Conclusions

The contribution of drayage trucking activities to pollution circumstance around POI was investigated. During 2018, nearly 5350 tons of air pollutants were released. $NO_x$ was the principal source of air pollutants with a share of 69% of the total amount of emissions in tons, followed by CO with 18% of the total weight, while $SO_x$ and $NH_3$ emissions showed a minor share each. The contribution of PM emissions, including PM10 and PM2.5, are not serious in terms of weight, but the impact from them could be serious in practice due to their tiny sizes. In addition, Inner Port, South Port, and New Port contributed over 90% of the total amount of emissions in POI during 2018. The main reason here is that a huge amount of large and container trucks have been operated at those three ports. As mentioned above, large and container trucks are dominant sources of emissions in POI. Large and container trucks mostly use diesel fuel which emitted more emissions, compared to LPG.

Several recommendations about policies are suggested to regulate and control the negative impacts of drayage truck operation in POI on local sustainability.

### 5.1. The Establishment of Emission Control Area (ECA) and "Affected Zone"

The establishment of ECA not only on the sea but also on land at POI is necessary to reduce local air pollution to protect residential areas. Twenty-one percent of drayage trucks traveling in the POI area are manufactured before 2006, which have old generation and not environmental-friendly engines (Euro 0-III). The amount of $NO_x$ and CO emitted from these old engines is much higher than from new ones. Therefore, if the entrance and operation of trucks that use the old engine are limited or banned completely, it is expected that 50% amount of CO emission and $NO_x$ emission could be cut down. In practice, the ageing vehicle control system has been conducted in metropolitan cities in Korea since 2009. Especially, old vehicles made before 2005 are prohibited entering urban areas. However, due to the limitation in number of equipped cameras, the complexity of road system and the low perception of drivers, the system has not worked effectively. However, the limitation of entrance roads and port gates lights up the expectation of an efficient ageing vehicle control system in port gate area. The old-age drayage trucks will be prevented at port gates and they also may be fined for entering urban area.

The sulfur content in fuel also decides the amount of $SO_x$ and PM emissions released. Although the amount of $SO_x$ emission is small, if the requirement of sulfur content, especially in diesel fuel, is adjusted from 15 ppm to below 5 ppm by the International Maritime Organization (IMO), the amount of $SO_x$ and PM emissions also will be decreased significantly. Moreover, the regulation of using LPG, especially in ECA instead of diesel as fuel for drayage trucks, also helps to decrease the amount of $SO_x$ and PM emissions.

As mentioned above, this study analyzed PM emission in the buffer "affected zone" within 0.5 km from the port gate. Actually this zone does not exist, however, this zone should be established in practice. In this zone, population density should be stretched and kept at a low level to minimize negative impacts. Park might be placed in that area to absorb noise as well as emissions and prevent the flows of polluted air moving straight into the residential areas. In addition, methods to reduce PM concentration levels as watering road should be conducted frequently in this zone. Congestion conditions often happen in this zone, as results of the bottleneck in port gate, therefore, the infrastructure system in this zone also needs to be improved and expanded or re-scheduled during peak hours to reduce congestion.

*5.2. Optimization of Terminal Operation*

The goal of an efficient port is reaching a rational equilibrium between ship-to-shore, container yard, and on-land operation activities. Of course, this challenge is hard and complicated to seek in practice, resulting in the unreliable schedule of arriving vessels and spreading of trucks entering port to pick-up or deliver goods. As mentioned in literature review, there are two solutions that can help decrease urban freight traffic, then, improve environmental performance of port.

The first solution is reducing empty-truck trips by applying TAS. In term of port operators, TAS can provide truck arrival information, hence, operators can plan well and improve container handling operations. By reducing container re-handles, yard crane productivity can be improved significantly. In addition, it also helps reduce delay time in the port gate and terminal. TAS also offers cost-saving for truck drivers and shippers: Driver can earn more money with the same working time and shippers can reduce financial burden. After that, customers also can receive cheaper products due to cost reduction. Last but not least, limitation of empty trips also offers green benefits. A cut down in empty kilometers running leads to reduction in emissions emitted. Fewer trucks operating on the road also means less traffic, reducing congestion and accidents.

To implement TAS, POI firstly should decide a basic manner that fits its operational and commercial interests:

1. Appointment window: Appointment window states the period of time, within which driver with booked code can arrive at the entry gate. These windows should be set from 30 min up to a few hours. Due to congestion at port gate, trucks often join queue and arrival actual entry gate later.
2. Penalties: penalties system should be applied for late arrival. However, a grace period (10–15 min) should be considered.
3. The cut-off time: If the shipping schedule is fixed, the cut-off time should be set one day before the appointment. Then, both terminal operators and carriers have enough time for preparation.
4. To increasing commercial benefits, POI could consider peak-period appointment fees like Port of Vancouver.
5. Another option here is developing urban rail freight transportation system, that can combine with existing metro system. The metro system is developed in Korea, connected airport, seaport, cities, and dense residential areas. By using metro system, underground stations can replace rail freight station or big terminal to load and unload cargoes. In addition, cargoes could be delivered into underground station, then residents can pick-up their parcels on the way coming home or walk a short distance from their home to underground station. This reduces burden on last-mile delivery system, then improve traffic condition.

*5.3. Integrated Information Systems*

An integrated system should be set up to collect traffic data from ships and drayage trucks, then rebalance logistic demand and facility supply, or reschedule operation time to remove the bottleneck. However, the current data collection system and integrated system need to be upgraded to support the above goals. The data collection system is equipped only in port gate with limited

camera quantity to count daily number of entranced drayage trucks and finite vehicle characteristics like truck types, license plates, but not to record other in-port activities and status of trucks. In addition, the system should be linked with other transport-related data sources like Korea Transportation Safe Authority to officially collect vehicle characteristics data including manufacturing year, engine model, displacement, etc. Also, it is interesting that there is the existing of vehicles not registered officially. This encumbered and delayed the data collection or data smoothing process. It is suggested that the cooperation between POI and local authority is required to identify and capture operating illegal vehicles.

Besides estimating emissions from ports, it is essential to consider the dispersion of air pollution to the local community in the integrated system. In addition, the number of PM emissions from port could affect PM concentrations and the population-weighted mean of PM in areas surrounding port. The seasonal and daily variation of air pollutants concentration should be integrated with meteorological data and the number of emissions from port. Then, based on these variations, port operation activities would be re-planned promptly to minimize or keep the negative impacts on the local community at a safe level.

There are several benefits from TASs, however, they also can reduce the flexibility of supply chain. Strict time and insufficient slots require them to manage time slots to move containers. This is expected to motivate truck-sharing option among carriers. However, the lack of information, resources and trust between carriers constrains the effective use of TAS. Therefore, POI with the role of terminal operator should operate a platform where POI can provide necessary information about slots in terminal, shipping schedule, etc., and carriers also can find reliable truck-sharing options. Thus, this platform could motivate proper collaboration among carriers, then, enhance further efficiencies of TAS.

Last but not least, average or grouping inputs (traveling speed and distance, cargo and truck weight, meteorological data, etc.) applied in the calculation process might lead to unexpected uncertainty due to lacking detailed data. Therefore, an integrated information system collecting and managing big data, which combines information from the data collection stations or equipment placed in and around POI, is suggested. If these big data are linked directly to the integrated information system and the necessary data set is created, port-related truck emissions that affect urban area negatively can be estimated immediately and used for port operation properly.

**Author Contributions:** This paper represents the results of teamwork. All of the authors jointly designed the research methodology and carried out the case study.

**Funding:** This research was funded by Incheon National University Research Grant in 2019.

**Conflicts of Interest:** The authors declare no conflicts of interest.

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
