# Peer review of "A Study on Emissions from Drayage Trucks in the Port City-Focusing on the Port of Incheon"

_sustainability, doi:10.3390/su11195358_

Round 1
Reviewer 1 Report
This is a concise and well-presented paper. It calculated the pollutant emissions from trucks in ports and discussed the related negative environmental impacts as well as potential policy solutions.
I would suggest that it would be better if the authors could provide a geographical visualized emission map to further assist their arguments in the discussion section.
Author Response
|
Comment(s) |
Response(s) |
|
it would be better if the authors could provide a geographical visualized emission map to further assist their arguments in the discussion section. |
Geographical visualized emission and population density map was added in part 3.1. |
Reviewer 2 Report
Thanks the editor to invite me to review this paper.
The paper is interesting and well written here are some suggestions to improve the paper quality:
- in my opinion "estimation" presented as the aim of the study in the abstract is a little narrow. It must be some solution established as the aim. Also the results presented n the abstract are too detailed showing that the analysis is the only aim - it must be improved
- there is some literaturę review part in the introduction but it should be separate section
- there is no discussion related to other studies in this area !
- very weak recommendations to the practice
- it seems to me that the paper is not finished … it ends on the analysis, only
Author Response
|
Comment(s) |
Response(s) |
|
in my opinion "estimation" presented as the aim of the study in the abstract is a little narrow. It must be some solution established as the aim. Also the results presented n the abstract are too detailed showing that the analysis is the only aim - it must be improved |
Abstract was edited based on your comments. |
|
there is some literaturę review part in the introduction but it should be separate section |
Introduction part was re-organized and Literature Review part was separated. |
|
there is no discussion related to other studies in this area |
More studies were added and discussed. |
|
very weak recommendations to the practice |
New recommendations were added in a practical point of view. |
Reviewer 3 Report
The paper is very interesting and presents very important issue regarding environmental degradation caused by trucks. The title of the paper is adequate to the content of the paper.
However I have some suggestions and remarks which could improve the value of the paper:
Lines 40-44, The problem of freight transport planning has been widely discussed by Maria Lindholm, which publications are missing in the references.
Line 49-50, “In terms of social impact, freight transport is considered as the main contributor to urban traffic issues, especially congestion.” This sentence is not clear. What does it mean social impact? Social impact of freight transport? In the literature social impact of transport is more related to the safety problem. Congestion can be considered as both a social problem and economic problem.
Lines 55-57, “Thanks to increasing understanding about environmental impacts, local authorities and academic fields accepted the significant contribution of urban freight transport to air pollutant emissions, not only greenhouse gases but also non-greenhouse gases including particulate matter (PM) and nitrogen oxides”. This sentence is not grammatically correct. “…understanding about environmental impacts…” (on what?), “local authorities and academic fields accepted…” (What does it mean they accepted environmental impact?? – they became interested in this problem? All local authorities and academicians ?).
What does it mean “significant contribution of urban freight transport to air pollutant emissions..”? What about the impact on air pollution by passenger transport?
Lines - 58-59, “In France, urban freight transport was responsible…” Please provide the year to which provided information on NOx and PM emission is refereeing to. Is it 2010?
Line - 60, the data for London refers to 2010 year. In the database of Eurostat much more updated data can be found. Please update data in the paper.
Line 61 – are trains diesel vehicles? I think that today trains are mostly electric but maybe I am wrong.
Lines 122-124, justification of the subject of the article should refer more to the scientific (not only research) gap - for example to the development of a new research method.
Lines 125-127, “Therefore, this paper aims to clarify the significant contribution of trucking activities in the Incheon port area to urban air pollution, then, suggest and discuss possible policies that might help cut down serious emissions”. How do you know that this impact is significant on the urban pollution? And how can you assess the significance? I think that the purpose should be more thoroughly clarified.
Line 128 - annual paved road dust (PM) emission- do you mean Particulate Matter emission? – It is not only road dust.
Section 2.1. , The scope of the research should be more detailed described. Please shortly characterize each port, what are the flows of trucks and ships to particular ports, how far from the city centre is located each port, maybe it is also good to present it on the geographical map.
Line 145-146, - “…entering and leaving..”? That's mean that data includes the total number of entering and leaving trucks or total number of trucks which entered the port (and of course have to leave it later), please explain data collection more clearly.
Lines 148-149, - on the basis of which source did you classify trucks (small less than 1 ton of goods, medium 1-5 tones and large more than 5 tons)? Why private cars and passenger buses are not presented in the table? If you do not focused on them why are they mentioned (line 148)? The same question refers to information mentioned in the paper (lines 150-151) that buses are only operating cruises? Is this information necessary if you do not consider passenger transport?
Line 155 “Emission standards are based on…” is this enough source for emission calculation?
Table 2, On the basis of which source did you classify trucks (per production year)? According to which classification (EURO standards?) It could be worth to explain it in the text.
Table 4, I think it could be worth to present data (introduced in table 2 and 3) broken down into each analysed port.
The results should be described in more detail. It would increase the value of the paper if the authors went a little deeper into research results. This section in current form is more like report than scientific work.
266-267, “The remarkable contribution of drayage trucking activities to pollution circumstance around POI was clarified and highlighted”. - How can you assess if it is remarkable or not? How does it look like in other ports? What is the total air pollution (regarding various pollutant) in the analysed city? This conclusion is too general.
line 296-297, “It is agreed that reducing the number of trucks and waiting time at gate can decrease amount of emissions from trucks”. Did you calculated it? It is obvious conclusion, so it could be worth to show how much it can be reduced pollutant emissions from trucks (in value or percentage).
Line 303-305, “POI should limit the empty truck trips by motivating collaboration between drivers, logistics companies and shipping lines. Forming alliances could help reduce load imbalances and reduce costs. Empty trips also could be cut down by conducting the Truck appointment systems (TASs), which support to truck-sharing options”. How can it be done? Are there any examples of this solution in the world? Can you describe them?
Minor mistakes:
Line 73, “On the other hand”, where is the first hand ? You should avoid this type of expressions.
Line 117-120, “As a result, local authority in Incheon, home of the second largest port in Korea, has to face a dilemma: on the positive side, international trade and increasing logistics activities will offer more jobs and contribute actively to local economic development and competitiveness; however, larger negative impacts also will come together with benefits and affect seriously to life quality of local citizen”, this sentence is too long and hard to get through. Please split it into two or three short sentences.
Table 3. The title is rather wrong “by manufacturing year”, in my opinion it should be: “by fuel types”.
Author Response
|
Comment(s) |
Response(s) |
|
Lines 40-44, The problem of freight transport planning has been widely discussed by Maria Lindholm, which publications are missing in the references |
Literature was added and updataed. |
|
Line 49-50, “In terms of social impact, freight transport is considered as the main contributor to urban traffic issues, especially congestion.” This sentence is not clear. What does it mean social impact? Social impact of freight transport? In the literature social impact of transport is more related to the safety problem. Congestion can be considered as both a social problem and economic problem. |
“In terms of social impact” was removed. The authors discussed congestion as one of harmful effects of urban freight transport. |
|
Lines 55-57, “Thanks to increasing understanding about environmental impacts, local authorities and academic fields accepted the significant contribution of urban freight transport to air pollutant emissions, not only greenhouse gases but also non-greenhouse gases including particulate matter (PM) and nitrogen oxides”. This sentence is not grammatically correct. “…understanding about environmental impacts…” (on what?), “local authorities and academic fields accepted…” (What does it mean they accepted environmental impact?? – they became interested in this problem? All local authorities and academicians ?). |
The sentence was edited properly. |
|
What does it mean “significant contribution of urban freight transport to air pollutant emissions..”? What about the impact on air pollution by passenger transport? |
“significant” was removed in the sentance. The paper provided data on the number of trucks and the amount of emissions in the urban area from literature |
|
Lines - 58-59, “In France, urban freight transport was responsible…” Please provide the year to which provided information on NOx and PM emission is refereeing to. Is it 2010? |
This citation was replaced by another data in 2017. |
|
Line - 60, the data for London refers to 2010 year. In the database of Eurostat much more updated data can be found. Please update data in the paper. |
Data was updated. |
|
Line 61 – are trains diesel vehicles? I think that today trains are mostly electric but maybe I am wrong. |
In Incheon Port, diesel train is still used. |
|
Lines 122-124, justification of the subject of the article should refer more to the scientific (not only research) gap - for example to the development of a new research method. |
Literature review was updated. |
|
Lines 125-127, “Therefore, this paper aims to clarify the significant contribution of trucking activities in the Incheon port area to urban air pollution, then, suggest and discuss possible policies that might help cut down serious emissions”. How do you know that this impact is significant on the urban pollution? And how can you assess the significance? I think that the purpose should be more thoroughly clarified. |
“significant” was removed in the sentance. In the paper, “contribution to urban air pollution” implies the amount of emissions around POI. |
|
Line 128 - annual paved road dust (PM) emission- do you mean Particulate Matter emission? – It is not only road dust. |
“annual paved road dust (PM) emission” was corrected to “annual paved road particulate emission”. |
|
Section 2.1. , The scope of the research should be more detailed described. Please shortly characterize each port, what are the flows of trucks and ships to particular ports, how far from the city centre is located each port, maybe it is also good to present it on the geographical map. |
Description about each port and geographical map were added. |
|
Line 145-146, - “…entering and leaving..”? That's mean that data includes the total number of entering and leaving trucks or total number of trucks which entered the port (and of course have to leave it later), plaease explain data collection more clearly. |
It is total number of trucks which entered the port during 2018. |
|
Lines 148-149, - on the basis of which source did you classify trucks (small less than 1 ton of goods, medium 1-5 tones and large more than 5 tons)? Why private cars and passenger buses are not presented in the table? If you do not focused on them why are they mentioned (line 148)? The same question refers to information mentioned in the paper (lines 150-151) that buses are only operating cruises? Is this information necessary if you do not consider passenger transport? |
The authors catagories based on record data collected by POI, including 4 groups for drayage trucks: Small truck (less than 1 ton of goods), medium truck (carrying 1-5 tons of goods), large truck (more than 5 tons of goods), and container truck. Information about private car and bus were removed. |
|
Line 155 “Emission standards are based on…” is this enough source for emission calculation? |
Emission standards EURO 0~VI are defined based on year of vehicle production. Emission factor is decided considering EURO emission standards, type of fuel, and other vehicle information in the estimation process. |
|
Table 2, On the basis of which source did you classify trucks (per production year)? According to which classification (EURO standards?) It could be worth to explain it in the text. |
Trucks are classified based on EURO emission standards. Table 2 was edited on the basis of EURO standards.
|
|
I think it could be worth to present data (introduced in table 2 and 3) broken down into each analysed port. |
Table 2 and table 3 were edited. |
|
The results should be described in more detail. It would increase the value of the paper if the authors went a little deeper into research results. This section in current form is more like report than scientific work. |
The results are described in more detail. |
|
266-267, “The remarkable contribution of drayage trucking activities to pollution circumstance around POI was clarified and highlighted”. - How can you assess if it is remarkable or not? How does it look like in other ports? What is the total air pollution (regarding various pollutant) in the analysed city? This conclusion is too general. |
“Remarkable” was removed. More discussion on results was added Detaild guide line was added in Conclution.
|
|
line 296-297, “It is agreed that reducing the number of trucks and waiting time at gate can decrease amount of emissions from trucks”. Did you calculated it? It is obvious conclusion, so it could be worth to show how much it can be reduced pollutant emissions from trucks (in value or percentage). |
The sentence was replaced by: “Quantity of vehicle and operating time including idle time are main factors that determine amount of emission released, therefore, amount of emission reduced will be proportional to the number of trucks decreased or idle time cut down.” |
|
Line 303-305, “POI should limit the empty truck trips by motivating collaboration between drivers, logistics companies and shipping lines. Forming alliances could help reduce load imbalances and reduce costs. Empty trips also could be cut down by conducting the Truck appointment systems (TASs), which support to truck-sharing options”. How can it be done? Are there any examples of this solution in the world? Can you describe them? |
Description about TASs as well as examples were added into Discussion. |
|
Line 73, “On the other hand”, where is the first hand ? You should avoid this type of expressions |
“On the other hand” was removed. |
|
Line 117-120, “As a result, local authority in Incheon, home of the second largest port in Korea, has to face a dilemma: on the positive side, international trade and increasing logistics activities will offer more jobs and contribute actively to local economic development and competitiveness; however, larger negative impacts also will come together with benefits and affect seriously to life quality of local citizen”, this sentence is too long and hard to get through. Please split it into two or three short sentences. |
The sentence was splited to 3 short sentences. |
|
Table 3. The title is rather wrong “by manufacturing year”, in my opinion it should be: “by fuel types”. |
The title was corrected as “by fuel types”. |
Reviewer 4 Report
The paper addresses a very interesting and appealing research topic. The structure of the paper is correct. In order to enhance the quality of this paper, the following issues have to be addressed before considering the paper for the publication:
Although the main aspects related to drayage trucks has been investigated, please provide a more detail literature review about the existing models allow to reduce the trucks adoption by dry port implementation, with particular reference to case in which a rail transport is allows to directly connect the sea port with the dry port. Consistently with this purpose the literature review should be updated in order to enhance, broaden and deepen the state of the art. As a suggestion, the models developed in accordance with this approach has been addressed by: https://doi.org/10.24425/mper.2019.128240 https://doi.org/10.3390/su11061515 http://doi.org/10.1590/0103-6513.20170074 Please define the acronym of the existing models introduced in literature review (e.g. HBEFA, GREET, LEM, etc.); Many parameters showed in model description (equations from 2 to 9), are not defined (e.g. epsilon, beta and delta in eq. 4); Please detail in dept the definition and the possible values to be assigned to different parameters adopted by calculation process (section 2.3), for e.g. “RF” (pag. 5) is defined as “specific coefficients”, what means? In other cases, the numerical value assigned to parameter is given, but the parameter is not defined (e.g. “l” in pag. 6); Please check the use of subscripts in different equations, for e.g. in eq. 3 the parameter “efhot,k” is equal to parameter “efhot,i,k,r”, in eq. 5 are showed parameters “efhoti,,k” (with two commas) and “efhoti,k”, is the same parameter? The Dust (PM) emission form abrasion of paved road (Edust) and the emission factor for slit loading (EF) are introduced in equations 9 and 10, please detail how these factors affect the emission from drayage trucks defined in equation 1; Please justify the results showed in section 3, it is not clear how the authors are able to estimate the value of each air pollutants (e.g. CO, NOx, SOx, etc.) adopting the calculation process showed in previously section. The results of the model introduced allows to identify the total amount of drayage trucking emission as an aggregated information, or can be adopted for the evaluation of the each pollutant? Please check the follow mistakes: The font adopted in first part of section 3 is in bold and the text shall be written justified; The reference 28 in “References” section is not mentioned in text of the manuscript.
Author Response
|
Comment(s) |
Response(s) |
|
Although the main aspects related to drayage trucks has been investigated, please provide a more detail literature review about the existing models allow to reduce the trucks adoption by dry port implementation, with particular reference to case in which a rail transport is allows to directly connect the sea port with the dry port. Consistently with this purpose the literature review should be updated in order to enhance, broaden and deepen the state of the art. As a suggestion, the models developed in accordance with this approach has been addressed by: https://doi.org/10.24425/mper.2019.128240 https://doi.org/10.3390/su11061515 http://doi.org/10.1590/0103-6513.20170074 |
Literature review was updated. |
|
Please define the acronym of the existing models introduced in literature review (e.g. HBEFA, GREET, LEM, etc.); |
Acronyms were defined. |
|
Many parameters showed in model description (equations from 2 to 9), are not defined (e.g. epsilon, beta and delta in eq. 4) In other cases, the numerical value assigned to parameter is given, but the parameter is not defined (e.g. “l” in pag. 6); |
Definitions of the parameters were added. |
|
Please detail in dept the definition and the possible values to be assigned to different parameters adopted by calculation process (section 2.3), for e.g. “RF” (pag. 5) is defined as “specific coefficients”, what means? |
RF is re-defined as “reduction factor”. |
|
Please check the use of subscripts in different equations, for e.g. in eq. 3 the parameter “efhot,k” is equal to parameter “efhot,i,k,r”, in eq. 5 are showed parameters “efhoti,,k” (with two commas) and “efhoti,k”, is the same parameter? |
Equations were edited properly. |
|
The Dust (PM) emission form abrasion of paved road (Edust) and the emission factor for slit loading (EF) are introduced in equations 9 and 10, please detail how these factors affect the emission from drayage trucks defined in equation 1 |
They are separated parts. Equation 1 is used for engine emission and equations 9,10 are used for paved road emission. Detail explaintation was added. |
|
Please justify the results showed in section 3, it is not clear how the authors are able to estimate the value of each air pollutants (e.g. CO, NOx, SOx, etc.) adopting the calculation process showed in previously section |
Exhaust emission is applied concurrently for all target pollutants. Fuel evaporation happens with hydrocarbon pollutants like VOC
PM tire and break emission is for PM emission
|
|
The font adopted in first part of section 3 is in bold and the text shall be written justified |
The text was checked and modified. |
|
The reference 28 in “References” section is not mentioned in text of the manuscript. |
The authors mentioned it in part 3.1. Research scope |
Round 2
Reviewer 2 Report
Dear Editor
The Authors have addressed almost all of the suggestions, please ask them to motivate why the paper is focus n analysis mainly
Author Response
There has been no effort to analyze emissions released from POI. The paper attempts to propose a proper emission estimation process applying the most suitable methodologies for POI. This paper provides a guideline for continuous calculation in the future, which can be useful information for green policy implications.
Reviewer 3 Report
The paper has been significantly improved and now, in my opinion, can be published in the journal.
Author Response

(The authors gave the same response as above.)
